# Vaccine effectiveness against severe COVID-19 outcomes within the French overseas territories: A cohort study of 2-doses vaccinated individuals matched to unvaccinated ones followed up until September 2021 and based on the National Health Data System

**Laura Semenzato**[1]*, **Jérémie Botton**[1], **Bérangère Baricault**[1], **Jacqueline Deloumeaux**[2], **Clarisse Joachim**[3], **Emmanuelle Sylvestre**[4], **Rosemary Dray-Spira**[1], **Alain Weill**[1], **Mahmoud Zureik**[1]

1 EPI-PHARE Scientific Interest Group in Epidemiology of Health Products (French National Agency for the Safety of Medicines and Health Products—ANSM, French National Health Insurance—CNAM), Saint-Denis, France, 2 Clinical Research and Innovation Department, Guadeloupe University Hospital Center, Les Abymes, Guadeloupe, 3 Martinique General Cancer Registry, Cancerology Hematology Urology Center from Martinique University Hospital Center, Fort-de-France, Martinique, 4 Clinical Data center—Martinique University Hospital Center, Fort-de-France, Martinique

* laura.semenzato@assurance-maladie.fr

## Abstract

### Importance

Although several observational studies on the effectiveness of SARS-CoV-2 vaccination have been published, vaccination coverage by August, 3 2021, remained low in the French overseas territories, despite Martinique and Guadeloupe experiencing an unprecedented number of COVID-19-related hospitalizations. We aimed to determine the association between COVID-19 vaccination and severe COVID-19 in the French overseas territories.

### Methods

The French National Health Data System was used to conduct a 1:1 matched-cohort study. For each individual receiving a first dose of BNT162b2, mRNA-1273, ChAdOx1 nCoV-19, or Ad26.COV2-S vaccine between December 27, 2020, and July 31, 2021, one unvaccinated individual was randomly selected and matched for year of birth, sex, and overseas territories on the date of vaccination. We estimated vaccine effectiveness against COVID-19-related hospitalization and in-hospital death after a full vaccination schedule, defined as ≥14 days after the second dose. Analyses were stratified according to the number of comorbidities.

### Results

276,778 vaccinated individuals had a double-dose vaccination during the follow-up period and were followed with their paired unvaccinated control. The average age was 50 years and 53% were women. During a median 77 days of follow-up from day 14 after the second

**Data Availability Statement:** According to data protection and the French regulation, the authors cannot publicly release the data from the French national health data system (SNDS). However, any person or structure, public or private, for-profit or nonprofit, is able to access SNDS data upon authorization from the French Data Protection Office (CNIL Commission Nationale de l'Informatique et des Libertés) to carry out a study, a research, or an evaluation of public interest (https://www.snds.gouv.fr/SNDS/Processus-d-acces-aux-donnees and https://www.indsante.fr/). Here is a non-author point of contact where data requests can be sent https://www.health-data-hub.fr/contact.

**Funding:** The author(s) received no specific funding for this work.

**Competing interests:** The authors have declared that no competing interests exist.

injection, 96 COVID-19-related hospitalizations occurred among vaccinated individuals and 1,465 among their unvaccinated counterparts. Overall, vaccine effectiveness against hospitalization was 94% (95%CI [93–95]) and exceeded 90% in each overseas territory, except Mayotte. The results were similar looking specifically at hospitalizations between July 15 and September 30, 2021. Vaccine effectiveness against in-hospital death was similar (94% [95%CI 91–96]). The risk of COVID-19-related hospitalization increased with the number of comorbidities, especially among vaccinated individuals.

## Conclusions and relevance

In conclusion, vaccination has a major effect in reducing the risk of severe Covid-19 in the French overseas territories. The risk of COVID-19-hospitalization was very low among vaccinated individuals, especially in the absence of comorbidities. These results aim to increase confidence in vaccine effectiveness in overseas territories in hope of achieving better vaccination coverage.

## Introduction

Vaccine efficacy [1–4] in clinical trials against symptomatic COVID-19 has ranged from 70 to 95% and real-world effectiveness of the mRNA and ChAdOx1 nCoV-19 vaccines [5–12] against COVID-19 hospitalization was estimated to be approximately 90%. In France, where 66.6 million people reside in mainland France and 2.2 million in French overseas territories, the vaccination campaign began on December 27, 2020, and was extended to all adults in mid-May 2021 and to children aged 12 and older in mid-June. However, vaccination coverage by August 3, 2021 remained low in the French overseas territories: approximately 43% of patients had received at least one injection in Reunion (Indian Ocean) and 20–25% in Mayotte (Indian Ocean) and the overseas territories in the Caribbean (Martinique, Guadeloupe) and South America (French Guiana) versus 65% in mainland France [13]. Moreover, viral circulation differed between mainland France and the French overseas territories. In mainland France in 2021, a high number of COVID-19-related hospitalizations were first observed in March and April, followed by a lower increase of cases in July and August, in the context of high vaccination coverage. Martinique and Guadeloupe and, to a lesser extent Reunion and French Guiana, experienced an unprecedented number of COVID-19-related hospitalizations since the beginning of the epidemic between mid-July and the end of September 2021, related to the emergence of the Delta variant. We estimated vaccine effectiveness against severe COVID-19 in the French overseas territories over a period including that of high viral circulation.

## Materials and methods

We used the French National Health Data System (Système National des Données de Santé, SNDS) to conduct a 1:1 matched-cohort study (more details in the eMethods in S1 File). For each individual receiving a first dose of one of the first vaccines authorized in France (i.e., BNT162b2, mRNA-1273, ChAdOx1 nCoV-19, or Ad26.COV2.S) between December 27, 2020, and July 31, 2021, one unvaccinated control matched for year of birth, sex, and overseas territories, was randomly selected on the date of vaccination (first injection). If a control subject was vaccinated during follow-up, follow-up was stopped for the pair, and the subject was then eligible to be included as a vaccinated subject. We chose not to exclude the few patients

vaccinated with the Ad26.COV2.S vaccine whose vaccination schedule was first in a single injection before requiring a booster dose from late summer. We estimated vaccine effectiveness after a full vaccination schedule, defined as ≥14 days after the second dose, as the percentage of risk reduction (1-Hazard Ratio) in COVID-19-related hospitalization and in-hospital COVID-19-related death using Cox proportional hazards models adjusted for the 7 sociodemographic characteristics, history of COVID-19 and the 23 comorbidities. Each subject was followed until COVID-19-related hospitalization, death, or the end of follow-up on September 30, 2021. If one individual of a matched pair died or had an event of interest, the other individual was not censored. Additional pair-censoring analyses on death or event of at least one individual in the matched pair were performed in S1 Table. We performed sensitivity analyses: first, we focused on events occurring over a period where the delta variant was predominant, from July 15 2021, to September 30, 2021, by excluding pairs with at least one individual hospitalized before July 15, 2021. Then, we excluded pairs with at least one individual having COVID-19 history (S2 Table). We also investigated effectiveness according to the number of comorbidities, defined as a categorical variable ranging from 0–5 or more, based on the sum of 47 identified chronic conditions (S3 Table) [14].

### Ethic statement

The research group has permanent regulatory access to the data from the French National Health Data System (French decree No. 2016–1871 of December 26, 2016, on the processing of personal data called National Health Data System and French law articles Art. R. 1461–13 and 14) *upon authorization from the French Data Protection Office (CNIL Commission Nationale de l'Informatique et des Libertés)*. No informed consent was required because the data are anonymized.

### Results and discussion

In total, 276,778 individuals (97.8% [n = 270,641] with the BNT162b2, 0.9% [n = 2,581] with the mRNA-1273, 1.3% [n = 3,505] with the ChAdOx1 nCoV-19, and 0.0% [n = 8] with the Ad26.COV2.S vaccine) had a double-dose vaccination during the follow-up period (patients 'distribution by month of 2$^{nd}$ injection in S4 Table) and were followed with their unvaccinated counterparts. Supplementary analysis restricted to BNT162b2 2-doses-vaccinated individuals during the follow-up period showed no difference in outcome (S5 Table). Among vaccinated individuals, 46,221 lived in Guadeloupe, 43,264 in Martinique, 26,328 in French Guiana, 149,683 in Reunion, and 11,282 in Mayotte. They were, on average, 50 years old and 53% were women (Table 1). The two groups were relatively similar in terms of health characteristics.

During a median follow-up of 77 days (interquartile range, 42–111) from day 14 after the second injection, 96 COVID-19-related hospitalizations occurred among vaccinated and 1,465 among unvaccinated individuals. 45% and 40% respectively had a SARS-COV-2 positive test, which occurred on average 5 days before hospitalization. Overall vaccine effectiveness against hospitalization was estimated to be 94% (95%CI [93–95]). It exceeded 90% in each overseas territory, except Mayotte, where it was 78% (Table 2). Vaccine effectiveness against in-hospital death was similar when adjusting for all the variables (Table 2) or when restricting to a few ones (S6 Table): 94% (95%CI [91–96]), with 19 deaths among vaccinated and 285 among unvaccinated individuals. Similar results were also obtained when restricting the study period from July 15 2021, to September 30, 2021, only (Table 3) or when excluding pairs with at least one individual having COVID-19 history (S2 Table).

Among hospitalized patients, 26% of those vaccinated had at least five comorbidities versus only 7% among unvaccinated patients. COVID-19-related hospitalizations increased with the

**Table 1. Characteristics of the study population with double-dose vaccination and their paired unvaccinated individuals.**

| Baseline characteristics | | No vaccine (n = 276,778) | Vaccine (n = 276,778) | Standardized differences |
|---|---|---|---|---|
| Age (year) | Mean (SD) | 49.81 (18.88) | 49.81 (18.88) | 0.00000 |
| Age | -20 | 19,610 (7.1) | 19,610 (7.1) | |
| | 20–29 | 27,665 (10.0) | 27,665 (10.0) | |
| | 30–39 | 37,412 (13.5) | 37,412 (13.5) | |
| | 40–49 | 46,048 (16.6) | 46,048 (16.6) | |
| | 50–54 | 27,207 (9.8) | 27,207 (9.8) | |
| | 55–59 | 28,507 (10.3) | 28,507 (10.3) | |
| | 60–64 | 25,036 (9.0) | 25,036 (9.0) | |
| | 65–69 | 21,864 (7.9) | 21,864 (7.9) | |
| | 70–74 | 16,555 (6.0) | 16,555 (6.0) | |
| | 75–79 | 11,774 (4.3) | 11,774 (4.3) | |
| | 80–84 | 7,875 (2.8) | 7,875 (2.8) | |
| | 85–89 | 4,645 (1.7) | 4,645 (1.7) | |
| | 90+ | 2,580 (0.9) | 2,580 (0.9) | |
| Sex | Women | 147,452 (53.3) | 147,452 (53.3) | 0.00000 |
| | Men | 129,326 (46.7) | 129,326 (46.7) | |
| Administrative regions | Guadeloupe | 46,221 (16.7) | 46,221 (16.7) | 0.00000 |
| | French Guiana | 26,328 (9.5) | 26,328 (9.5) | |
| | Reunion Island | 149,683 (54.1) | 149,683 (54.1) | |
| | Martinique | 43,264 (15.6) | 43,264 (15.6) | |
| | Mayotte | 11,282 (4.1) | 11,282 (4.1) | |
| Flu vaccination in 2018 or 2019 | No | 274,147 (99.0) | 269,355 (97.3) | 0.12992 |
| | Yes | 2,631 (1.0) | 7,423 (2.7) | |
| Frailty | No | 257,031 (92.9) | 261,338 (94.4) | -0.06381 |
| | Yes | 19,747 (7.1) | 15,440 (5.6) | |
| Alcoholism | No | 272,853 (98.6) | 273,974 (99.0) | -0.03697 |
| | Yes | 3,925 (1.4) | 2,804 (1.0) | |
| Smoking | No | 270,945 (97.9) | 268,593 (97.0) | 0.05411 |
| | Yes | 5,833 (2.1) | 8,185 (3.0) | |
| Hypertension | No | 216,952 (78.4) | 209,894 (75.8) | 0.06073 |
| | Yes | 59,826 (21.6) | 66,884 (24.2) | |
| Diabetes | No | 244,852 (88.5) | 242,626 (87.7) | 0.02481 |
| | Yes | 31,926 (11.5) | 34,152 (12.3) | |
| Dyslipidemia | No | 253,739 (91.7) | 245,879 (88.8) | 0.09587 |
| | Yes | 23,039 (8.3) | 30,899 (11.2) | |
| Coronary diseases | No | 271,250 (98.0) | 269,569 (97.4) | 0.04052 |
| | Yes | 5,528 (2.0) | 7,209 (2.6) | |
| Heart failure | No | 273,728 (98.9) | 274,117 (99.0) | -0.01391 |
| | Yes | 3,050 (1.1) | 2,661 (1.0) | |
| Cardiac rhythm or conduction disturbances | No | 272,811 (98.6) | 272,457 (98.4) | 0.01053 |
| | Yes | 3,967 (1.4) | 4,321 (1.6) | |
| Valvular diseases | No | 274,927 (99.3) | 274,718 (99.3) | 0.00902 |
| | Yes | 1,851 (0.7) | 2,060 (0.7) | |
| Obliterating arterial disease of the lower limbs | No | 274,167 (99.1) | 274,064 (99.0) | 0.00381 |
| | Yes | 2,611 (0.9) | 2,714 (1.0) | |
| Stroke | No | 271,155 (98.0) | 272,111 (98.3) | -0.02557 |

(*Continued*)

**Table 1.** (Continued)

| Baseline characteristics | | No vaccine (n = 276,778) | Vaccine (n = 276,778) | Standardized differences |
|---|---|---|---|---|
| | Yes | 5,623 (2.0) | 4,667 (1.7) | |
| Pulmonary embolism | No | 276,199 (99.8) | 276,238 (99.8) | -0.00314 |
| | Yes | 579 (0.2) | 540 (0.2) | |
| Chronic respiratory diseases (cystic fibrosis excluded) | No | 264,813 (95.7) | 262,000 (94.7) | 0.04741 |
| | Yes | 11,965 (4.3) | 14,778 (5.3) | |
| Chronic dialyses | No | 276,304 (99.8) | 275,945 (99.7) | 0.02673 |
| | Yes | 474 (0.2) | 833 (0.3) | |
| Renal transplant | No | 276,628 (99.9) | 276,425 (99.9) | 0.02434 |
| | Yes | 150 (0.1) | 353 (0.1) | |
| Liver diseases | No | 275,038 (99.4) | 274,872 (99.3) | 0.00741 |
| | Yes | 1,740 (0.6) | 1,906 (0.7) | |
| Active cancer | No | 273,118 (98.7) | 271,895 (98.2) | 0.03585 |
| | Yes | 3,660 (1.3) | 4,883 (1.8) | |
| Neurotic or Mood Disorders, use of antidepressants | No | 269,651 (97.4) | 266,757 (96.4) | 0.06038 |
| | Yes | 7,127 (2.6) | 10,021 (3.6) | |
| Psychotic disorders, use of neuroleptics | No | 272,385 (98.4) | 273,209 (98.7) | -0.02501 |
| | Yes | 4,393 (1.6) | 3,569 (1.3) | |
| Dementia (including Alzheimer's disease) | No | 274,519 (99.2) | 274,923 (99.3) | -0.01700 |
| | Yes | 2,259 (0.8) | 1,855 (0.7) | |
| Epilepsy | No | 274,786 (99.3) | 275,328 (99.5) | -0.02491 |
| | Yes | 1,992 (0.7) | 1,450 (0.5) | |
| Parkinson disease | No | 275,683 (99.6) | 275,838 (99.7) | -0.00925 |
| | Yes | 1,095 (0.4) | 940 (0.3) | |
| Chronic inflammatory bowel diseases | No | 276,203 (99.8) | 275,874 (99.7) | 0.02303 |
| | Yes | 575 (0.2) | 904 (0.3) | |
| Rheumatoid arthritis and related diseases | No | 275,792 (99.6) | 275,438 (99.5) | 0.01977 |
| | Yes | 986 (0.4) | 1,340 (0.5) | |
| Ankylosing spondylitis and related diseases | No | 276,357 (99.8) | 276,070 (99.7) | 0.02299 |
| | Yes | 421 (0.2) | 708 (0.3) | |
| History of coronavirus infection (hospitalization or positive screening test) | No | 265,744 (96.0) | 275,120 (99.4) | -0.22779 |
| | Yes | 11,034 (4.0) | 1,658 (0.6) | |
| Type of COVID-19 vaccine administered during first injection | Pfizer | | 270,641 (97.8) | |
| | Moderna | | 2,581 (0.9) | |
| | Astrazeneca | | 3,505 (1.3) | |
| | Janssen | | 8 (0.0) | |

number of comorbidities, especially among vaccinated individuals (compared to patients without comorbidity, aHR 29.5 95%CI [13.4–64.9] among vaccinated with 5+ comorbidities and aHR 4.4 95%CI [3.5–5.6] among unvaccinated individuals with 5+ comorbidities) (Table 4). Vaccines were all the more effective among individuals with no or few comorbidities (VE of 97% [95%; 98%] among individuals with no comorbidity vs 83% [73%; 89%] among 5+) (Table 5).

Vaccines were highly effective against severe COVID-19 in the French overseas territories, as observed in mainland France. The lower effectiveness observed in Mayotte is likely related to the low number of events due to low viral circulation in Mayotte during this period, leading

**Table 2. Vaccine effectiveness measured as overall reduction of the risks of Covid-19-related hospitalization and in-hospital death from day 14 after the 2nd injection.** Hazard ratios (HRs) were obtained using Cox models taking into account all the variables described in Table 1.

| Criteria | Region | Vaccine exposition | Number of subjects | Number of event (%) | Median follow-up [interquartile range] | Crude HR (95% CI) | Adjusted HR (95% CI) | Vaccine effectiveness |
|---|---|---|---|---|---|---|---|---|
| Hospitalization | Overseas territories | no | 276778 | 1465 (0.53%) | 77 [42–111] | 1 | 1 | - |
| | | yes | 276778 | 96 (0.03%) | 77 [42–111] | 0.06 (0.05–0.08) | 0.06 (0.05–0.07) | 94% (93%; 95%) |
| | Guadeloupe | no | 46221 | 406 (0.88%) | 87 [53–120] | 1 | 1 | - |
| | | yes | 46221 | 46 (0.1%) | 87 [54–120] | 0.11 (0.08–0.15) | 0.09 (0.07–0.13) | 91% (87%; 93%) |
| | Martinique | no | 43264 | 488 (1.13%) | 94 [48–128] | 1 | 1 | - |
| | | yes | 43264 | 25 (0.06%) | 94 [48–129] | 0.05 (0.03–0.08) | 0.05 (0.03–0.07) | 95% (93%; 97%) |
| | French Guiana | no | 26328 | 147 (0.56%) | 98 [61–128] | 1 | 1 | - |
| | | yes | 26328 | 6 (0.02%) | 99 [62–129] | 0.04 (0.02–0.09) | 0.04 (0.02–0.09) | 96% (91%; 98%) |
| | Reunion Island | no | 149683 | 414 (0.28%) | 67 [38–97] | 1 | 1 | - |
| | | yes | 149683 | 16 (0.01%) | 68 [38–97] | 0.04 (0.02–0.06) | 0.04 (0.02–0.06) | 96% (94%; 98%) |
| | Mayotte | no | 11282 | 10 (0.09%) | 78 [44–114] | 1 | 1 | - |
| | | yes | 11282 | 3 (0.03%) | 78 [44–115] | 0.30 (0.08–1.08) | 0.22 (0.06–0.89) | 78% (11%; 94%) |
| Death | Overseas territories | no | 276778 | 285 (0.1%) | 77 [42–111] | 1 | 1 | - |
| | | yes | 276778 | 19 (0.01%) | 77 [42–111] | 0.07 (0.04–0.10) | 0.06 (0.04–0.09) | 94% (91%; 96%) |
| | Guadeloupe | no | 46221 | 96 (0.21%) | 87 [53–120] | 1 | 1 | - |
| | | yes | 46221 | 11 (0.02%) | 87 [54–120] | 0.11 (0.06–0.21) | 0.09 (0.05–0.17) | 91% (83%; 95%) |
| | Martinique | no | 43264 | 98 (0.23%) | 94 [48–128] | 1 | 1 | - |
| | | yes | 43264 | 2 (0%) | 94 [48–129] | 0.02 (0.00–0.08) | 0.02 (0.00–0.07) | 98% (93%; 100%) |
| | French Guiana | no | 26328 | 18 (0.07%) | 98 [61–128] | 1 | 1 | - |
| | | yes | 26328 | 1 (0%) | 99 [62–129] | 0.05 (0.01–0.41) | 0.04 (0.01–0.35) | 96% (65%; 99%) |
| | Reunion Island | no | 149683 | 70 (0.05%) | 67 [38–97] | 1 | 1 | - |
| | | yes | 149683 | 5 (0%) | 68 [38–97] | 0.07 (0.03–0.17) | 0.07 (0.03–0.17) | 93% (83%; 97%) |
| | Mayotte | no | 11282 | 3 (0.03%) | 78 [44–114] | 1 | 1 | - |
| | | yes | 11282 | . (.%) | 78 [44–115] | 0.00 (0.00 -.) | 0.00 (0.00 -.) | 100% (.%; 100%) |

to a much higher uncertainty for the estimates. The vaccine effectiveness appeared to persist during the strong epidemic of the Delta variant, which particularly affected Martinique and Guadeloupe, between mid-July and the end of September 2021, with more than 2,700 and 2,000 patients hospitalized for COVID-19, respectively, and the number of COVID-19 hospital deaths corresponding to 75% and 60%, respectively, of all COVID-19 hospital deaths registered in each of these territories between the beginning of the epidemic and the end of December 2021.

The overseas territories benefit from the mandatory French National Health system in the same way as mainland France. Despite the availability of vaccine doses within the same timeframe as in mainland France, utilization rates of doses delivered to overseas territories remained low. Such low vaccination coverage has been associated with a distrust in public

**Table 3. Vaccine effectiveness measured as overall reduction of the risk of Covid-19-related in-hospital death from day 14 after the 2nd injection for hospitalizations from July 15 to September 30, 2021, only.** Hazard ratio (HRs) were obtained using Cox models taking into account all the variables described in Table 1.

| Region | Vaccine exposition | Number of subjects | Number of event (%) | Median follow-up [interquartile range] | Crude HR (95% CI) | Adjusted HR (95% CI) | Vaccine effectiveness |
|---|---|---|---|---|---|---|---|
| Overseas territories | no | 276,778 | 1,206 (0.44%) | 78 [42–111] | 1 | 1 | - |
| | yes | 276,778 | 79 (0.03%) | 78 [42–111] | 0.07 (0.05–0.08) | 0.06 (0.05–0.07) | 94% (93%; 95%) |
| Guadeloupe | no | 46,221 | 375 (0.82%) | 87 [54–120] | 1 | 1 | - |
| | yes | 46,221 | 43 (0.09%) | 87.5 [54–120] | 0.11 (0.08–0.16) | 0.09 (0.07–0.13) | 91% (87%; 93%) |
| Martinique | no | 43,264 | 462 (1.07%) | 94 [48–129] | 1 | 1 | - |
| | yes | 43,264 | 23 (0.05%) | 95 [49–129] | 0.05 (0.03–0.08) | 0.05 (0.03–0.07) | 95% (93%; 97%) |
| French Guiana | no | 26,328 | 75 (0.29%) | 100 [63–129] | 1 | 1 | - |
| | yes | 26,328 | 2 (0.01%) | 100 [63–129] | 0.03 (0.01–0.11) | 0.03 (0.01–0.11) | 97% (89%; 99%) |
| Reunion Island | no | 149,683 | 289 (0.2%) | 68 [38–97] | 1 | 1 | - |
| | yes | 149,683 | 9 (0.01%) | 68 [38–97] | 0.03 (0.02–0.06) | 0.03 (0.01–0.06) | 97% (94%; 99%) |
| Mayotte | no | 11,282 | 5 (0.05%) | 79 [44–119] | 1 | 1 | - |
| | yes | 11,282 | 2 (0.02%) | 79 [44–119] | 0.40 (0.08–2.06) | 0.36 (0.07–1.92) | 64% (-92%; 93%) |

authority related to complex and multifactorial issues [15], including the management of chlordecone use, a persistent organochlorine pesticide, prohibited in the US since 1976, in France since 1990, and since 1993 in the French West Indies. Low vaccination rates have also been observed in Montserrat and the British Virgin Islands, British overseas territories also located in the Caribbean. However, other overseas territories benefit from high vaccination coverage, such as most of the British Overseas Territories, with some of the highest global vaccination rates [16], or Puerto Rico since the decision of strict vaccine mandates guided by accurate scientific results in the real world on its population [17].

The stronger association among 2-doses-vaccinated individuals between the residual risk of COVID-19-related hospitalization and the number of comorbidities should not be interpreted as vaccine failures. Vaccines were highly effective, but even more so among those with no or few comorbidities (Table 5). This is of particular interest, as people from the French overseas territories have higher frequencies of diabetes, chronic dialysis, and stoke than in mainland France [18], which are medical risk factors for the development of severe COVID-19 [14].

**Table 4. Distribution, hazard ratios (HRs), and 95% confidence intervals (95%CIs) for COVID-19-related hospitalization and in-hospital mortality according to exposure and the number of comorbidities.**

| | Number | | | | Number of events | | | | Model adjusted for age and sex only | | Multivariable model | |
|---|---|---|---|---|---|---|---|---|---|---|---|---|
| | Vaccinated | | Unvaccinated | | | | Vaccinated | | Unvaccinated | | Vaccinated | Unvaccinated |
| **Vaccinated** | | | | | | | | | Unvaccinated | | | |
| **Number of comorbidities** | **276,778** | | **276,778** | | **96** | **%** | **1,465** | **%** | | | | |
| 0 | 168,704 | 61% | 181,083 | 65% | 12 | 13% | 458 | 31% | 1 | 1 | 1 | 1 |
| 1 | 50,546 | 18% | 46,258 | 17% | 11 | 11% | 309 | 21% | 1.81 (0.78–4.20) | 1.74 (1.50–2.02) | 1.86 (0.80–4.33) | 1.69 (1.45–1.96) |
| 2 | 26,411 | 10% | 23,202 | 8% | 18 | 19% | 261 | 18% | 4.64 (2.12–10.14) | 2.43 (2.06–2.86) | 4.87 (2.22–10.71) | 2.40 (2.03–2.83) |
| 3 | 17,447 | 6% | 14,627 | 5% | 15 | 16% | 224 | 15% | 5.40 (2.37–12.31) | 3.01 (2.53–3.58) | 5.86 (2.56–13.45) | 3.05 (2.56–3.63) |
| 4 | 8,153 | 3% | 6,818 | 2% | 15 | 16% | 107 | 7% | 10.34 (4.50–23.74) | 2.94 (2.35–3.68) | 11.75 (5.08–27.14) | 3.12 (2.49–3.91) |
| 5 or more | 5,517 | 2% | 4,790 | 2% | 25 | 26% | 106 | 7% | 23.90 (10.99–51.96) | 3.93 (3.13–4.94) | 29.52 (13.42–64.91) | 4.44 (3.53–5.58) |

**Table 5. Vaccine effectiveness measured as overall reduction of the risk of Covid-19-related hospitalization from day 14 after the 2nd injection stratified by the number of comorbidities.**

| Criteria | Number of comorbidities | Vaccine exposition | Number of subjects | Number of event (%) | Median follow-up [interquartile range] | Crude HR (95% CI) | Adjusted HR (95% CI) | Vaccine effectiveness |
|---|---|---|---|---|---|---|---|---|
| Hospitalization | 0 | no | 181083 | 458 (0.25%) | 70 [40–102] | 1 | 1 | - |
| | | yes | 168704 | 12 (0.01%) | 68 [38–99] | 0.03 (0.02–0.05) | 0.03 (0.02–0.05) | 97% (95%; 98%) |
| | 1 | no | 46258 | 309 (0.67%) | 85 [48–118] | 1 | 1 | - |
| | | yes | 50546 | 11 (0.02%) | 85 [48–119] | 0.03 (0.02–0.06) | 0.03 (0.02–0.06) | 97% (94%; 98%) |
| | 2 | no | 23202 | 261 (1.12%) | 91 [54–125] | 1 | 1 | - |
| | | yes | 26411 | 18 (0.07%) | 94 [57–127] | 0.06 (0.04–0.10) | 0.06 (0.04–0.10) | 94% (90%; 96%) |
| | 3 | no | 14627 | 224 (1.53%) | 95 [57–129] | 1 | 1 | - |
| | | yes | 17447 | 15 (0.09%) | 98 [62–132] | 0.06 (0.03–0.09) | 0.05 (0.03–0.09) | 95% (91%; 97%) |
| | 4 | no | 6818 | 107 (1.57%) | 97 [58–132] | 1 | 1 | - |
| | | yes | 8153 | 15 (0.18%) | 101 [66–140] | 0.11 (0.06–0.19) | 0.10 (0.05–0.17) | 90% (83%; 95%) |
| | 5 or more | no | 4790 | 106 (2.21%) | 99 [61–134] | 1 | 1 | - |
| | | yes | 5517 | 25 (0.45%) | 106 [70–143] | 0.18 (0.12–0.28) | 0.17 (0.11–0.27) | 83% (73%; 89%) |

Although vaccination coverage has improved in the French overseas territories, it remains much lower than in mainland France: in mid-February 2022, more than 78% of the mainland French population had received at least one injection and 57% had had a booster injection versus 51% and 24%, respectively, in the overseas territories.

The SNDS is a comprehensive claims database that allowed us to analyse vaccine effectiveness for individuals from the whole population of the French overseas territories benefiting from a 2-dose vaccination schedule, thus limiting selection bias. We focused on patients requiring admission specifically for SARS-CoV-2 infection, and due to the invoicing nature of the codifications recorded by public and private healthcare establishments, diagnostic coding errors are likely to be rare. Given the predominance of the Delta variant over the period analyzed, our estimates may not apply to periods of high circulation of other variants, such as Omicron. We may have a misclassification bias due to differential health care seeking behavior between vaccinated individuals and unvaccinated controls. This may lead to a potential underestimation of illness among unvaccinated individuals and then an underestimation of vaccine effectiveness. Health care seeking behavior may also be considered as a confounding factor: people with less use of healthcare would be less vaccinated and hospitalized for covid-19. However, this would have a limited impact due to the severity of the endpoint, and would then lead to an underestimation of vaccine effectiveness. Our study may also have been affected by other residual confounding due to differences between vaccinated individuals and unvaccinated controls after matching and adjustment for a high range of comorbidities that were shown to be associated with COVID-19-related hospitalization.

In conclusion, vaccines had a major effect in reducing COVID-19-related hospitalizations in the French overseas territories. The risk of COVID-19-hospitalization was very low among vaccinated individuals, especially in the absence of comorbidities. Achieving better vaccination coverage in overseas territories would limit the incidence of severe COVID-19.

## Supporting information

**S1 Table. Vaccine effectiveness against COVID-19-related hospitalisation depending on censoring variable.**
(DOCX)

**S2 Table. Vaccine effectiveness measured as overall reduction of the risk of Covid-19-related hospitalization from day 14 after the 2nd injection for individuals without COVID-19 history.** Hazard ratios (HRs) were obtained using Cox models taking into account all the variables described in Table 1.
(DOCX)

**S3 Table. Algorithms for identification of chonic conditions.**
(DOCX)

**S4 Table. Patients 'distribution by month of 2nd injection.**
(DOCX)

**S5 Table. Vaccine effectiveness measured as overall reduction of the risk of Covid-19-related hospitalization from day 14 after the 2nd injection for individuals fully vaccinated with the BNT162b2 vaccine.** Hazard ratios (HRs) were obtained using Cox models taking into account all the variables described in Table 1.
(DOCX)

**S6 Table. Vaccine effectiveness measured as overall reduction of the risk of Covid-19-related and in-hospital death from day 14 after the 2nd injection.** Hazard ratios (HRs) were obtained using Cox models taking into account age ($<75$ vs $\geq75$), gender, combining alcoholism and smoking, frailty and number of comorbidity.
(DOCX)

**S1 File.**
(DOCX)

## Author Contributions

**Conceptualization:** Laura Semenzato, Jérémie Botton, Bérangère Baricault, Rosemary Dray-Spira, Alain Weill, Mahmoud Zureik.

**Data curation:** Laura Semenzato.

**Formal analysis:** Laura Semenzato.

**Methodology:** Laura Semenzato, Jérémie Botton.

**Supervision:** Rosemary Dray-Spira, Alain Weill, Mahmoud Zureik.

**Writing – original draft:** Laura Semenzato.

**Writing – review & editing:** Jérémie Botton, Jacqueline Deloumeaux, Clarisse Joachim, Emmanuelle Sylvestre, Rosemary Dray-Spira, Alain Weill, Mahmoud Zureik.

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
