## [Decision Letter · Decision Letter 0]

30 May 2022

PONE-D-22-10419Vaccine effectiveness against severe COVID-19 outcomes within the French overseas territories: a cohort study based on the National Health Data System.PLOS ONE

Dear Dr. Semenzato,

Thank you for submitting your manuscript to PLOS ONE. After careful consideration, we feel that it has merit but does not fully meet PLOS ONE’s publication criteria as it currently stands. Therefore, we invite you to submit a revised version of the manuscript that addresses the points raised during the review process.

 Both reviewers agree that your manuscript needs some clarifications changes and made many constructive suggestions. Please address all of these before resubmitting.

We look forward to receiving your revised manuscript.

Kind regards,

Joël Mossong, PhD

Academic Editor

PLOS ONE

Journal Requirements:

Reviewers' comments:

Reviewer's Responses to Questions

**Comments to the Author**

1. Is the manuscript technically sound, and do the data support the conclusions?

Reviewer #1: Yes

Reviewer #2: Yes

2. Has the statistical analysis been performed appropriately and rigorously? 

Reviewer #1: I Don't Know

Reviewer #2: Yes

3. Have the authors made all data underlying the findings in their manuscript fully available?

Reviewer #1: Yes

Reviewer #2: Yes

4. Is the manuscript presented in an intelligible fashion and written in standard English?

Reviewer #1: Yes

Reviewer #2: Yes

5. Review Comments to the Author

Reviewer #1: This manuscript presents a matched cohort study based on data from the French National Health Data System. The main finding is that two Covid-19 vaccinations have been highly effective against severe Covid-19 outcomes in 2021 in the French overseas territories. This is important to note as the vaccination coverage in these regions was surprisingly low. Region-specific might help (might already have helped) raising the vaccine uptake and benefiting public health.

Major comments:

- The length of the study (median follow-up 77 days) perfectly suits the purpose of demonstrating the benefit of vaccination in 2021 in order to raise vaccination coverage levels. However, since then new variants have emerged, and additional vaccine doses have been broadly recommended. I therefore suggest giving a reference to either the study period or the studied vaccine schedule (2 doses) and length of follow-up in the title.

- It is unclear how the matched pair was treated if the unvaccinated individual received a vaccination during follow-up. This needs to be explained.

- The sensitivity analysis which focused on the delta period should be described better. 1. Mention it in the methods section. 2. I assume that it was not only the time period for case/event recruitment that was restricted to July – December but it was the whole study period/follow-up that was restricted. This should be stated correctly and clearly in the abstract (lines 19-20).

- Results, lines 17-18: This is unclear. The number of deaths in the text are smaller than any count given in Table 4.

- Table 2: % risk reduction is the effect measure that is in the main text referred to as vaccine effectiveness. Please clearly define VE as % risk reduction in the methods and then also mention the term vaccine effectiveness here in the table. Why does the interquartile range of the median follow-up differ between the vaccinated and unvaccinated. Is it because if one individuals of a matched pair died, the other individual was not censored? If so, this might lead to bias. Please state how the variables in Table 1 were taken into account.

- There are 3 pages of covariates and other than that some of them were used for matching, it is unclear how the others were used. In the methods covariate adjustment is mentioned. But were all the distinct comorbidities considered separately or were they somehow combined? I doubt the study size and case numbers were big enough to allow for that many covariates.

- Table 3: Is there something missing, is the table legend wrong or were the outcomes hospitalization and mortality combined in this analysis? The outline of the results suggest that you run two models, one for the vaccinated and one for the unvaccinated. Is that correct? In this case you allowed the effect of age and sex to be different in the two groups. Why? Such an effect is not considered in the main VE analysis, is it? It would be great to add the stratified VE estimates, i.e. VE in the group without comorbidities, VE in the group with exactly 1 comorbidity etc. Like this you could demonstrate the benefit of vaccination depending on the number of comorbidities.

- Discussion, line 22: This has not been shown. You need to show the VE stratified by the number of comorbidities, or you rephrase the second part of the sentence.

Minor comments:

- The outcome or events of interest are hospitalization or death. The effect measure is vaccine effectiveness against these outcomes (not against the risk). Please correct this in the abstract and elsewhere, where you write a) that the outcome was vaccine effectiveness and b) that you estimated vaccine effectiveness against the risk of hospitalization/death. The table captions where you refer to the reduction of the risk are perfectly fine though.

- Please carefully review the terminology and use the terms consistently: e.g., vaccine effectiveness vs. vaccination effectiveness; Guiana vs. Guyana

- Abstract, line 13: Split the sentence into two sentences and clearly state how number of comorbidities was taken into account.

- Replace “ more significantly” by other phrases that would express that the effect in vaccinated is stronger

- Choose either “Reference” or “1” for the ratio estimates of the reference group

Reviewer #2: Dear Editors of Plos One and authors of manuscript,

Semenzato et al present vaccine effectiveness estimates from French overseas territories. The manuscript is well written and the analyzes is conducted in good manner. However, I would like to point out some improvement:

Major comments:

1) I would like to see in a figure what is the time since the second dose among the vaccinated (for example the N of two dose vaccinated in every month).

2) How is the area of residency defined? Is this municipality based or just the overseas territories? I don’t understand this from

3) Were the individuals with prior SARS-CoV-2 infection to 27th Dec included in the study? They should be excluded or somehow taken into account in the analysis (adjusting or stratification).

4) How is the unvaccinated defined? Unvaccinated in the end of follow-up or at the time of matching? This is unclear.

5) I would like to see a table of comorbidities used in table 3. If I understood correctly, they are defined in prior article, but it is laborious to check this in prior article.

6) What was the time from positive SARS-CoV-2 sample to Covid-19 hospitalization? Were all hospitalized or died persons tested for SARS-CoV-2 infection?

7) I don’t understand how the matching for age was done. In the manuscript it is written “2021, one unvaccinated control matched for year of birth” and in supplements “Age was defined as a categorical variable by five-year-age groups”. Was the age matched with each birth year or +/- 5 years from birth?

Minor comments:

1) Results: ”We chose not to exclude the few patients vaccinated with the Ad26.COV2.S vaccine whose vaccination schedule was first in a single injection before requiring a booster dose from late summer.” This is methods.

2) In the supplements it is written ” We used the social deprivation index as a measure of socioeconomic status. This indicator is based on the median household income, the percentage of high school graduates in the population over the age of 15, the percentage of manual workers in the labor force, and the unemployment rate for the person's town of residence.” Which part did the authors used this social deprivation index? It is unclear.

3) Table 3: There is quite significant difference of risk among vaccinated with 5 or more comorbidities compared to unvaccinated as the authors discussed. Have the authors checked if number of comorbidities is higher among vaccinated (if vaccinated had for example 8 comorbidities on average compared 6 among unvaccinated)?

6. PLOS authors have the option to publish the peer review history of their article (what does this mean?). If published, this will include your full peer review and any attached files.

Reviewer #1: No

Reviewer #2: No

---

## [Author Response · Author response to Decision Letter 0]

15 Jun 2022

Reviewer #1: This manuscript presents a matched cohort study based on data from the French National Health Data System. The main finding is that two Covid-19 vaccinations have been highly effective against severe Covid-19 outcomes in 2021 in the French overseas territories. This is important to note as the vaccination coverage in these regions was surprisingly low. Region-specific might help (might already have helped) raising the vaccine uptake and benefiting public health.

We thank the reviewer for the assessment of our manuscript and highlighting the importance of this work.

Major comments:

- The length of the study (median follow-up 77 days) perfectly suits the purpose of demonstrating the benefit of vaccination in 2021 in order to raise vaccination coverage levels. However, since then new variants have emerged, and additional vaccine doses have been broadly recommended. I therefore suggest giving a reference to either the study period or the studied vaccine schedule (2 doses) and length of follow-up in the title.

We have enriched the title as follows: “Vaccine effectiveness against severe COVID-19 outcomes within the French overseas territories: a cohort study of 2-doses matched vaccinated individuals until September 2021 based on the National Health Data System”.

- It is unclear how the matched pair was treated if the unvaccinated individual received a vaccination during follow-up. This needs to be explained.

We agree that this important information was missing in the manuscript for proper understanding. 

We added this explanation on page 4 line 23.

‘If a control subject was vaccinated during follow-up, follow-up was stopped for the pair, and the subject was then eligible to be included as a vaccinated subject.’

- The sensitivity analysis which focused on the delta period should be described better. 1. Mention it in the methods section. 2. I assume that it was not only the time period for case/event recruitment that was restricted to July – December but it was the whole study period/follow-up that was restricted. This should be stated correctly and clearly in the abstract (lines 19-20).

In this sensitivity analysis, we considered all pairs without any hospitalisation before July 15, 2021 and focused on events occurring on the delta period, e.g. from July 15, 2021 to September, 30 2021.

In the abstract on page 2 line 23 we specified:

“The results were similar when looking specifically at hospitalizations between July 15 and September 30, 2021”

In the Material and Methods section, we added on page 5 line 7:

“We performed a sensitivity analysis focusing on events occurring over a period where the delta variant was predominant, from July 15 2021, to September 30, 2021, excluding pairs with at least one individual hospitalized before July 15, 2021.”

- Results, lines 17-18: This is unclear. The number of deaths in the text are smaller than any count given in Table 4.

We were referring in this sentence two 2 distinct analyses presented separately in two tables, which was indeed not clear. 

We now specify in the revised version of the manuscript on page 5 line 27 that information concerning the number of deaths was presented in Table 2 while the results limited to the period where the delta variant was predominant are presented in Table 3.

- Table 2: % risk reduction is the effect measure that is in the main text referred to as vaccine effectiveness. Please clearly define VE as % risk reduction in the methods and then also mention the term vaccine effectiveness here in the table. Why does the interquartile range of the median follow-up differ between the vaccinated and unvaccinated. Is it because if one individuals of a matched pair died, the other individual was not censored? If so, this might lead to bias. Please state how the variables in Table 1 were taken into account.

We specified in the Material and Methods section on page 4 line 27:

“We estimated vaccine effectiveness after a full vaccination schedule, defined as ≥14 days after the second dose, as the percentage of risk reduction (1-Hazard Ratio) in COVID-19-related hospitalization and in-hospital COVID-19-related death using Cox proportional hazards models adjusted for baseline characteristics and comorbidities.”

Regarding the second point raised, indeed, if one individual of a matched pair died or had an event of interest, the other individual was not censored, which explains slight differences in interquartile range between vaccinated and unvaccinated individuals. We performed additional pair-censoring analyses on death or event of at least one individual in the matched pair. Similar results were obtained and added in the supplementary materials in Table S1.

Table S1: Vaccine effectiveness against COVID-19-related hospitalisation depending on censoring variable.

Model Vaccine exposition Number of subjects Number of event (%) Median follow-up [interquartile range] Crude HR Adjusted HR % risk reduction

 (95% CI) (95% CI) 

0 no 276778 1465 (0.53%) 77 [42 - 111] Reference Reference -

 yes 276778 96 (0.03%) 77 [42 - 111] 0.06 (0.05 - 0.08) 0.06 (0.05 - 0.07) 94% (93% ; 95%)

1 no 276778 1459 (0.53%) 77 [42 - 111] Reference Reference -

 yes 276778 95 (0.03%) 77 [42 - 111] 0.06 (0.05 - 0.08) 0.06 (0.05 - 0.07) 94% (93% ; 95%)

2 no 276778 1459 (0.53%) 77 [42 - 111] Reference Reference -

 yes 276778 93 (0.03%) 77 [42 - 111] 0.06 (0.05 - 0.08) 0.06 (0.05 - 0.07) 94% (93% ; 95%)

Model 0: reference model presented in the manuscript: if one individual of the pair died or had an event of interest, i.e. hospitalisation or in-hospital death, the pair was not censored.

Model 1: model with pair-censoring on death (if one individual of the pair died, the pair was censored).

Model 2: model with pair-censoring on death or event of interest (if one individual of the pair died or had an event of interest, i.e. hospitalisation or in-hospital death, the pair was censored).

We added on page 5 line 4:

‘If one individual of a matched pair died or had an event of interest, the other individual was not censored. Additional pair-censoring analyses on death or event of at least one individual in the matched pair were performed in supplementary Table S1.’

How the variables in Table 1 were taken into account is explained in the answer to the next question.

- There are 3 pages of covariates and other than that some of them were used for matching, it is unclear how the others were used. In the methods covariate adjustment is mentioned. But were all the distinct comorbidities considered separately or were they somehow combined? I doubt the study size and case numbers were big enough to allow for that many covariates.

Each individual receiving a first dose of vaccine was matched for year of birth, sex, and overseas territories to one unvaccinated control, randomly selected on the date of vaccination. Then we adjusted simultaneously for the 7 sociodemographic characteristics, history of COVID-19 and the 23 comorbidities presented in the Table 1. We can notice in Table 2 the stability of the models with few differences between crude associations and fully adjusted models. 

We specified in the Methods section on page 4 line 27:

“We estimated vaccine effectiveness after a full vaccination schedule, defined as ≥14 days after the second dose, as the percentage of risk reduction (1-Hazard Ratio) in COVID-19-related hospitalization and in-hospital COVID-19-related death using Cox proportional hazards models adjusted for the 7 sociodemographic characteristics, history of COVID-19 and the 23 comorbidities presented in the Table 1”

- Table 3: Is there something missing, is the table legend wrong or were the outcomes hospitalization and mortality combined in this analysis? The outline of the results suggest that you run two models, one for the vaccinated and one for the unvaccinated. Is that correct? In this case you allowed the effect of age and sex to be different in the two groups. Why? Such an effect is not considered in the main VE analysis, is it? It would be great to add the stratified VE estimates, i.e. VE in the group without comorbidities, VE in the group with exactly 1 comorbidity etc. Like this you could demonstrate the benefit of vaccination depending on the number of comorbidities.

We indeed performed two models, one for the vaccinated and one for the unvaccinated, to estimate reduction risk of hospitalisations according to the number of comorbidities. 

As another way to examine the impact of the interaction between the number of comorbidities and the vaccination on COVID-19-related hospitalisation, we performed additional analyses of vaccine effectiveness stratified according to the number of comorbidities. Results are presented below and added in Table 5.

We can notice that vaccine effectiveness against COVID-19-related hospitalisations decreases with increasing number of comorbidity from 97% (95% ; 98%) for patients without any comorbidity to 83% (73% ; 89%) for patients with at least 5 comorbidities.

We added a reference to the Table 5 on page 5 line 27:

‘The risk of COVID-19-related hospitalization increased with the number of comorbidities, especially among vaccinated individuals (compared to patients without comorbidity, aHR 29.5 95%CI [13.4-64.9] among vaccinated with 5+ comorbidities and aHR 4.4 95%CI [3.5-5.6] among unvaccinated individuals with 5+ comorbidities) (Table 4). Vaccines were all the more effective among individuals with no or few comorbidities (VE of 97% [95%; 98%] among individuals with no comorbidity vs 83% [73%; 89%] among 5+) (Table 5).’

Table 5: Vaccine effectiveness against COVID-19-related hospitalisation by number of comorbidity.

Number of comorbidities Vaccine exposition Number of subjects Number of event (%) Median follow-up [interquartile range] Crude HR Adjusted HR % risk reduction

 (95% CI) (95% CI) 

0 no 181083 458 (0.25%) 70 [40 - 102] 1 1 -

 yes 168704 12 (0.01%) 68 [38 - 99] 0.03 (0.02 - 0.05) 0.03 (0.02 - 0.05) 97% (95% ; 98%)

1 no 46258 309 (0.67%) 85 [48 - 118] 1 1 -

 yes 50546 11 (0.02%) 85 [48 - 119] 0.03 (0.02 - 0.06) 0.03 (0.02 - 0.06) 97% (94% ; 98%)

2 no 23202 261 (1.12%) 91 [54 - 125] 1 1 -

 yes 26411 18 (0.07%) 94 [57 - 127] 0.06 (0.04 - 0.10) 0.06 (0.04 - 0.10) 94% (90% ; 96%)

3 no 14627 224 (1.53%) 95 [57 - 129] 1 1 -

 yes 17447 15 (0.09%) 98 [62 - 132] 0.06 (0.03 - 0.09) 0.05 (0.03 - 0.09) 95% (91% ; 97%)

4 no 6818 107 (1.57%) 97 [58 - 132] 1 1 -

 yes 8153 15 (0.18%) 101 [66 - 140] 0.11 (0.06 - 0.19) 0.10 (0.05 - 0.17) 90% (83% ; 95%)

5 no 4790 106 (2.21%) 99 [61 - 134] 1 1 -

 yes 5517 25 (0.45%) 106 [70 - 143] 0.18 (0.12 - 0.28) 0.17 (0.11 - 0.27) 83% (73% ; 89%)

- Discussion, line 22: This has not been shown. You need to show the VE stratified by the number of comorbidities, or you rephrase the second part of the sentence.

We added a reference to Table 5 describing VE stratified by the number of comorbidities on page 7 line 3:

‘Vaccines were highly effective, but even more so among those with no or few comorbidities (Table 5)’

Minor comments:

- The outcome or events of interest are hospitalization or death. The effect measure is vaccine effectiveness against these outcomes (not against the risk). Please correct this in the abstract and elsewhere, where you write a) that the outcome was vaccine effectiveness and b) that you estimated vaccine effectiveness against the risk of hospitalization/death. The table captions where you refer to the reduction of the risk are perfectly fine though.

We corrected in the Method section of the Abstract the sentence as follows: 

‘Events of interest were COVID-19-related hospitalization and in-hospital death. We then estimated vaccine effectiveness against COVID-19-related hospitalization after a full vaccination schedule, defined as ≥14 days after the second dose, and against in-hospital COVID-19-related death.’

Several ‘against the risk of’ were suppressed in the manuscript.

- Please carefully review the terminology and use the terms consistently: e.g., vaccine effectiveness vs. vaccination effectiveness; Guiana vs. Guyana

We corrected in the manuscript the specific terms mentioned.

- Abstract, line 13: Split the sentence into two sentences and clearly state how number of comorbidities was taken into account.

We specified in the Methods section of the Abstract:

‘We then estimated vaccine effectiveness against COVID-19-related hospitalization after a full vaccination schedule, defined as ≥14 days after the second dose, and against in-hospital COVID-19-related death. Finally, we estimated the risk of COVID-19-related hospitalization among vaccinated and unvaccinated individuals respectively, and vaccine effectiveness stratified according to the number of comorbidities, defined as a categorical variable ranging from 0-5 or more, based on the sum of 47 identified chronic conditions.’

- Replace “ more significantly” by other phrases that would express that the effect in vaccinated is stronger

We propose to replace “more significantly” by “especially” on page 5 line 28:

‘The risk of COVID-19-related hospitalization increased with the number of comorbidities, especially among vaccinated individuals.’

- Choose either “Reference” or “1” for the ratio estimates of the reference group

We choose 1 for the ratio estimates of the reference group. 

Reviewer #2: Dear Editors of Plos One and authors of manuscript,

Semenzato et al present vaccine effectiveness estimates from French overseas territories. The manuscript is well written and the analyzes is conducted in good manner. However, I would like to point out some improvement:

We thank the reviewer for the evaluation of our manuscript and for the advice.

Major comments:

1) I would like to see in a figure what is the time since the second dose among the vaccinated (for example the N of two dose vaccinated in every month).

You will find below the distribution of patients according to the month of full-vaccination. We added the table in Supplementary material in Table S3

Month of 2nd-injection Number of patients

January 99

February 4,853

March 12,249

April 20,630

May 41,248

June 67,417

July 57,605

August 72,674

September 3

We corrected in the Abstract and the Results section from ‘In total, 276,778 individuals had a double-dose vaccination by July, 31 (Table S3) and were followed with their unvaccinated counterparts.’ to ‘In total, 276,778 vaccinated individuals had a double-dose vaccination during the follow-up period and were followed with their paired unvaccinated control’ and added a reference to the table on page 5 line 14.

2) How is the area of residency defined? Is this municipality based or just the overseas territories? I don’t understand this from

We adjusted analyses for overseas territories; we specified this in the manuscript.

3) Were the individuals with prior SARS-CoV-2 infection to 27th Dec included in the study? They should be excluded or somehow taken into account in the analysis (adjusting or stratification).

Vaccination against COVID-19 started in December, 27th 2020 in France, that’s why individuals with prior infection to this date were not included in the study. 

You will find in the Methods section of the Manuscript the restriction of the inclusion period to patients receiving a first vaccine injection ‘between December 27, 2020, and July 31, 2021’.

4) How is the unvaccinated defined? Unvaccinated in the end of follow-up or at the time of matching? This is unclear.

‘For each individual receiving a first dose of COVID-19 vaccine between December 27, 2020, and July 31, 2021, one unvaccinated control matched for year of birth, sex, and overseas territories, was randomly selected on the date of vaccination (first injection). ‘ (on page 4 line 20)

An unvaccinated control was then found at the time of matching. 

For more clarification, we added the following sentence on page 4 line 27:

‘If a control subject was vaccinated during follow-up, follow-up was stopped for the pair, and the subject was then eligible to be included as a vaccinated subject.’

5) I would like to see a table of comorbidities used in table 3. If I understood correctly, they are defined in prior article, but it is laborious to check this in prior article.

We added in the Method section on page 5 line 8 a reference to the supplementary Table S2 containing algorithms for the identification of chronic pathologies used in the calculation of the number of comorbidities.

6) What was the time from positive SARS-CoV-2 sample to Covid-19 hospitalization? Were all hospitalized or died persons tested for SARS-CoV-2 infection?

Only 755/1,561 individuals (48%) had a positive SARS-CoV-2 test registered in the SIDEP database before or at the time of hospitalization: 712/1,465 (49%) among unvaccinated individuals and 43/1,465 (45%) among vaccinated individuals. This figure were of 113/285 (40%) and 9/19 (47%) for patients unvaccinated and vaccinated who died at hospital respectively.

For patients with a positive SARS-CoV-2 test registered in the SIDEP database before or at the time of hospitalization, the test was performed on average in the 5 days preceding hospitalisation (interquartile range 2 – 7 days). 

We added on page 5 line 24:

‘45% and 40% respectively had a SARS-COV-2 positive test, which occurred on average 5 days before hospitalization.’

7) I don’t understand how the matching for age was done. In the manuscript it is written “2021, one unvaccinated control matched for year of birth” and in supplements “Age was defined as a categorical variable by five-year-age groups”. Was the age matched with each birth year or +/- 5 years from birth?

For each individual receiving a first dose of COVID-19 vaccine between December 27, 2020, and July 31, 2021, one unvaccinated control matched for year of birth, sex, and overseas territories, was randomly selected on the date of vaccination (first injection). In Cox proportional hazards models adjusted for all baseline characteristics and comorbidities, we considered age as a categorical variable by five-year-age groups.

We specified in supplementary material on page 3 line 2:

‘We considered the patient's year of birth, gender, and overseas territories as matching variables. In adjusted Cox proportional hazards models, age was defined as a categorical variable by five-year-age groups’

Minor comments:

1) Results: ”We chose not to exclude the few patients vaccinated with the Ad26.COV2.S vaccine whose vaccination schedule was first in a single injection before requiring a booster dose from late summer.” This is methods.

We moved this sentence in the Method section.

2) In the supplements it is written ” We used the social deprivation index as a measure of socioeconomic status. This indicator is based on the median household income, the percentage of high school graduates in the population over the age of 15, the percentage of manual workers in the labor force, and the unemployment rate for the person's town of residence.” Which part did the authors used this social deprivation index? It is unclear.

Each individual receiving a first dose of vaccine was matched for year of birth, sex, and overseas territories to one unvaccinated control, randomly selected on the date of vaccination. Then we adjusted simultaneously for the 7 sociodemographic characteristics, history of COVID-19 and the 23 comorbidities presented in the Table 1. We can notice in Table 2 the stability of the models with few differences between crude associations and fully adjusted models. 

We specified in the Methods section on page 4 line 27:

“We estimated vaccine effectiveness after a full vaccination schedule, defined as ≥14 days after the second dose, as the percentage of risk reduction (1-Hazard Ratio) in COVID-19-related hospitalization and in-hospital COVID-19-related death using Cox proportional hazards models adjusted for the 7 sociodemographic characteristics, history of COVID-19 and the 23 comorbidities presented in the Table 1”

Social deprivation index was one of the several adjustment variables in Cox proportional hazards models.

We renamed the section in the supplements as ‘Description of the matching and adjustment variables‘. 

3) Table 3: There is quite significant difference of risk among vaccinated with 5 or more comorbidities compared to unvaccinated as the authors discussed. Have the authors checked if number of comorbidities is higher among vaccinated (if vaccinated had for example 8 comorbidities on average compared 6 among unvaccinated)?

We can already observe in Table 4 that the proportion of patients with much comorbidity is quite similar (slightly higher) between vaccinated and unvaccinated patients: 5% of vaccinated patients had 4 or more comorbidities vs 4% of unvaccinated patients.

The average number of comorbidity was 0.79 among vaccinated patients and 0.69 among unvaccinated patients. When we focused on patients with 5 or more comorbidities, there was no difference on the average number of comorbidities between vaccinated and unvaccinated individuals (5.63 vs 5.65, respectively).

---

## [Decision Letter · Decision Letter 1]

15 Jul 2022

PONE-D-22-10419R1Vaccine effectiveness against severe COVID-19 outcomes within the French overseas territories: a cohort study of 2-doses matched vaccinated individuals until September 2021 based on the National Health Data System.PLOS ONE

Dear Dr. Semenzato,

Thank you for submitting your manuscript to PLOS ONE. After careful consideration, we feel that it has merit but does not fully meet PLOS ONE’s publication criteria as it currently stands. Therefore, we invite you to submit a revised version of the manuscript that addresses the points raised during the review process.

ACADEMIC EDITOR:Both reviewers agree that your manuscript has improved but request still some minor changes.Please address these before resubmitting.

We look forward to receiving your revised manuscript.

Kind regards,

Joël Mossong, PhD

Academic Editor

PLOS ONE

Journal Requirements:

Reviewers' comments:

Reviewer's Responses to Questions

**Comments to the Author**

1. If the authors have adequately addressed your comments raised in a previous round of review and you feel that this manuscript is now acceptable for publication, you may indicate that here to bypass the “Comments to the Author” section, enter your conflict of interest statement in the “Confidential to Editor” section, and submit your "Accept" recommendation.

Reviewer #1: All comments have been addressed

Reviewer #2: All comments have been addressed

2. Is the manuscript technically sound, and do the data support the conclusions?

Reviewer #1: Yes

Reviewer #2: Yes

3. Has the statistical analysis been performed appropriately and rigorously? 

Reviewer #1: Yes

Reviewer #2: I Don't Know

4. Have the authors made all data underlying the findings in their manuscript fully available?

Reviewer #1: Yes

Reviewer #2: Yes

5. Is the manuscript presented in an intelligible fashion and written in standard English?

Reviewer #1: Yes

Reviewer #2: Yes

6. Review Comments to the Author

Reviewer #1: The methods section of the abstract could be slightly shortened: After listing the events of interest, they don’t have to be repeated in the next sentence about vaccine effectiveness. In addition, it’s enough to say that the analysis was stratified by number of comorbidities. No need to give the exact definition of number of comorbidities here in the abstract. As this will lower the word count, please bring back the phrase “the risk of” in the last sentence of the results section and the first two sentences of the conclusions. Also, in the main text “the risk of” should have not been deleted on page 36 lines 9-10, in the very last paragraph (conclusion), and the legends of table 2, 3, and 5. Table 5 is missing the “or more” phrase.

I didn’t find an explanation for why this specific set of 23 comorbidities was selected (“considered”) as covariates in the Cox model while the other available comorbidities were only counted in the overall number of comorbidities. The thought behind this selection should be explained. The comorbidities must have been selected a priori and not based on the data.

Indeed, covariate adjustment has almost no effect on the estimates. This should be briefly discussed. Is it because matching controlled for most of the differences between vaccinated and unvaccinated? The number of covariates is high. A rule of thumb states that 10-15 events are required per parameter to estimate (or control for) the parameter’s effect. There are alone 13 age groups (12 parameters) and 5 territories (4 parameters) in addition to more than 20 binary covariates (1 parameter each). Due to the high number of hospitalizations among the unvaccinated, the number of hospitalizations seems high enough for this complex model. However, the death counts are much lower. Also, many comorbidities or conditions (e.g. alcoholism) are very rare (<2%) in the cohort. Therefore, I really doubt that cohort size and incidence were high enough for this model. Please either discuss this problem as a potential limitation and/or run another sensitivity analysis (for the supplementary material) in which you don’t adjust for that many covariates by only using the number of comorbidities or using only the frailty indicator and perhaps by combining alcoholism and smoking (because adjusting for highly correlated covariates can negatively affect the variance of the estimates).

Last but not least, please briefly discuss the potential flaws of the data. Electronic health records strongly depend on health care seeking behavior and might be incomplete.

Reviewer #2: Dear editors and authors,

I thank Semenzato et al for providing the answers for my questions regarding their manuscript. The answers are good and comprehensive. I still have few minor comments:

1) Discussion: COVID-19 related hospitalization and the number of comorbidities should not be interpreted as vaccines failure.

a. I think ”vaccine failures” is correct

2) Tables 2, 3 and 5: Column ”% risk reduction”

a. In my mind it would be better to write ”vaccine effectiveness”

b. I would also name tables 3 and 5 like table 2 to harmonize the writing style� ”Vaccine effectiveness measured as overall…”.

3) What is the rationality to use 5 year age as a categorical variable by five-year-age groups in adjusting? Why not continuous variable?

a. However, I think this will not make huge difference for the results.

4) I understand from the answers that you exclude those with prior SARS-CoV-2 infection from the analysis. Is this mentioned in the manuscript?

5) The manuscript doesn’t have strengths and limitations part in the discussion. I would think that the manuscript would benefit from this. I would argue that one of the weakness is that all Covid-19 hospitalized did not have positive SARS-CoV-2 sample in the registers. One strength would be that you have quite comprehensive registers in use.

a. I’m sorry that I did not comment on this in the first revision round

7. PLOS authors have the option to publish the peer review history of their article (what does this mean?). If published, this will include your full peer review and any attached files.

Reviewer #1: No

Reviewer #2: No

---

## [Author Response · Author response to Decision Letter 1]

29 Jul 2022

Reviewer #1: The methods section of the abstract could be slightly shortened: After listing the events of interest, they don’t have to be repeated in the next sentence about vaccine effectiveness. In addition, it’s enough to say that the analysis was stratified by number of comorbidities. No need to give the exact definition of number of comorbidities here in the abstract. As this will lower the word count, please bring back the phrase “the risk of” in the last sentence of the results section and the first two sentences of the conclusions. Also, in the main text “the risk of” should have not been deleted on page 36 lines 9-10, in the very last paragraph (conclusion), and the legends of table 2, 3, and 5. Table 5 is missing the “or more” phrase.

We thank the reviewer for this second evaluation of our manuscript.

We changed the methods section of the abstract as follows:

‘The French National Health Data System was used to conduct a 1:1 matched-cohort study. For each individual receiving a first dose of BNT162b2, mRNA-1273, ChAdOx1 nCoV-19, or Ad26.COV2-S vaccine between December 27, 2020, and July 31, 2021, one unvaccinated individual was randomly selected and matched for year of birth, sex, and overseas territories on the date of vaccination. We estimated vaccine effectiveness against COVID-19-related hospitalization and in-hospital death after a full vaccination schedule, defined as ≥14 days after the second dose. Analyses were also stratified according to the number of comorbidities.’

As suggested, we bring back the phrase ‘the risk of’ in all the sentences/titles mentioned by the reviewer and we added ‘or more’ in Table 5.

I didn’t find an explanation for why this specific set of 23 comorbidities was selected (“considered”) as covariates in the Cox model while the other available comorbidities were only counted in the overall number of comorbidities. The thought behind this selection should be explained. The comorbidities must have been selected a priori and not based on the data.

We did not select comorbidities to include in the models based on the data. As you pointed out, the strategy adopted was different when the comorbidities were adjustment variables than when they were summarized in the variable number of comorbidities. Indeed, for example, Cox models were adjusted on active cancers whereas we distinguished breast, colorectal, lung, prostate and other cancers, active or passive, in the calculation of the number of comorbidities. It enabled us to limit the number of covariates to include in the Cox models, as pointed out in your next question. For the calculation of the number of comorbidities, there was no reason to reduce information a priori, and this would even be misleading because the number of comorbidities per person would have been underestimated.

We specified this on page 3 line 17 of the supplementary materials: 

‘The following 23 chronic diseases were considered in adjusted Cox models: cardiometabolic diseases, such as diabetes, hypertension, dyslipidaemia and/or lipid-lowering drug treatment or cardiovascular diseases (stroke and stroke sequelae, heart failure, coronary heart disease, cardiac arrhythmias or conduction disorders, valvular heart disease and peripheral artery disease), chronic respiratory diseases (excluding cystic fibrosis), pulmonary embolism, active cancers (female breast, lung, prostate, colorectal and other cancers), inflammatory diseases (chronic inflammatory bowel disease [IBD], rheumatoid arthritis, ankylosing spondylitis and related diseases), mental and behavioural disorders, neurodegenerative diseases, liver diseases, severe chronic kidney diseases and renal transplant. In the calculation of the number of comorbidities, we further distinguished breast, colorectal, lung, prostate and other active cancers from passive cancers, psoriasis, Down syndrome, multiple sclerosis, paraplegia, myopathy or myasthenia gravis, mental impairment, haemophilia or severe haemostasis disorders, HIV infections, cardiac, liver, lung transplants, i.e a total of 47 comorbidities.’

Indeed, covariate adjustment has almost no effect on the estimates. This should be briefly discussed. Is it because matching controlled for most of the differences between vaccinated and unvaccinated? The number of covariates is high. A rule of thumb states that 10-15 events are required per parameter to estimate (or control for) the parameter’s effect. There are alone 13 age groups (12 parameters) and 5 territories (4 parameters) in addition to more than 20 binary covariates (1 parameter each). Due to the high number of hospitalizations among the unvaccinated, the number of hospitalizations seems high enough for this complex model. However, the death counts are much lower. Also, many comorbidities or conditions (e.g. alcoholism) are very rare (<2%) in the cohort. Therefore, I really doubt that cohort size and incidence were high enough for this model. Please either discuss this problem as a potential limitation and/or run another sensitivity analysis (for the supplementary material) in which you don’t adjust for that many covariates by only using the number of comorbidities or using only the frailty indicator and perhaps by combining alcoholism and smoking (because adjusting for highly correlated covariates can negatively affect the variance of the estimates).

This is true that covariate adjustment had almost no effect on the estimates. Since age is a determinant of comorbidities and a strong risk factor of COVID-19, this is not so surprising that we obtained similar estimations between crude and fully adjusted model.

We had a low number of deaths among exposed and unexposed, 19 and 285 respectively. The reviewer is right that incidence is not high enough for most of the comorbidities in adjusted Cox model according to the rule of thumb. However, models seemed not be so unstable, probably because of the number of subjects overall. Nevertheless, we have performed a sensitivity analysis in which we only adjust on age (<75 vs ≥75), gender, combining alcoholism and smoking, frailty and number of comorbidity, and result obtained was similar (94% (90%; 96%) vs 94% (91%; 96%)).

S6 Table. Vaccine effectiveness measured as overall reduction of the risk of Covid-19-related and in-hospital death from day 14 after the 2nd injection. Hazard ratios (HRs) were obtained using Cox models taking into account age (<75 vs ≥75), gender, combining alcoholism and smoking, frailty and number of comorbidity.

Region Vaccine exposition Number of subjects Number of event (%) Median follow-up [interquartile range] Adjusted HR % risk reduction

 (95% CI) 

Overseas territories no 276778 285 (0.1%) 77 [42 - 111] 1 -

 yes 276778 19 (0.01%) 77 [42 - 111] 0.06 (0.04 - 0.10) 94% (90%; 96%)

We added on page 7 line 4:

‘Vaccine effectiveness against in-hospital death was similar when adjusting for all the variables (Table 2) or when restricting to a few ones (S6 Table): 94% (95%CI [91-96]), with 19 deaths among vaccinated and 285 among unvaccinated individuals; and when restricting the study period from July 15 2021, to September 30, 2021, only (Table 3).’

Last but not least, please briefly discuss the potential flaws of the data. Electronic health records strongly depend on health care seeking behavior and might be incomplete.

We added on page 8 line 28:

‘We may have a misclassification bias due to differential health care seeking behavior between vaccinated individuals and unvaccinated controls. This may lead to a potential underestimation of illness among unvaccinated individuals and then an underestimation of vaccine effectiveness. Health care seeking behavior may also be considered as a confounding factor: people with less use of healthcare would be less vaccinated and hospitalized for covid-19. However, this would have a limited impact due to the severity of the endpoint, and would then lead to an underestimation of vaccine effectiveness. Our study may also have been affected by other residual confounding due to differences between vaccinated individuals and unvaccinated controls after matching and adjustment for a high range of comorbidities that were shown to be associated with COVID-19-related hospitalization.’

Reviewer #2: Dear editors and authors,

We thank the reviewer for this second evaluation of our manuscript.

I thank Semenzato et al for providing the answers for my questions regarding their manuscript. The answers are good and comprehensive. I still have few minor comments:

1) Discussion: COVID-19 related hospitalization and the number of comorbidities should not be interpreted as vaccines failure.

a. I think ”vaccine failures” is correct

We made the proposed change.

2) Tables 2, 3 and 5: Column ”% risk reduction”

a. In my mind it would be better to write ”vaccine effectiveness”

b. I would also name tables 3 and 5 like table 2 to harmonize the writing style� ”Vaccine effectiveness measured as overall…”.

We homogenized the titles of the tables as suggested.

3) What is the rationality to use 5 year age as a categorical variable by five-year-age groups in adjusting? Why not continuous variable?

a. However, I think this will not make huge difference for the results.

We choose to use age as a as a categorical variable by five-year-age groups. This choice facilitates the description of the distribution of individuals, while keeping precise information, with age groups sufficiently restricted to avoid residual confusion but sufficiently broad to avoid a lack of subjects.

4) I understand from the answers that you exclude those with prior SARS-CoV-2 infection from the analysis. Is this mentioned in the manuscript?

I’m sorry, I misunderstood your question in the first revision and then gave you an incorrect answer.

We had already taken individuals with previous SARS-CoV-2 infection by adjusting for COVID-19 history in the Cox models (cf Tables 1 and 2), as you suggest in the first revision: 

‘Were the individuals with prior SARS-CoV-2 infection to 27th Dec included in the study? They should be excluded or somehow taken into account in the analysis (adjusting or stratification).’

We added here a sensitivity analysis on vaccine effectiveness measured as overall reduction of the risk of Covid-19-related hospitalization by excluding pairs of which at least one individual had a COVID-19 history and obtained similar results.

S2 Table: Vaccine effectiveness measured as overall reduction of the risk of Covid-19-related hospitalization from day 14 after the 2nd injection for individuals without COVID-19 history. Hazard ratios (HRs) were obtained using Cox models taking into account all the variables described in Table 1.

Region Vaccine exposition Number of subjects Number of event (%) Median follow-up [interquartile range] Adjusted HR % risk reduction

 (95% CI) 

Overseas territories no 264172 1444 (0.55%) 78 [42 - 111] 1 -

 yes 264172 91 (0.03%) 77 [43 - 112] 0.06 (0.05 - 0.07) 94% (93%; 95%)

We added on page 6 line 9:

‘Then, we excluded pairs with at least one individual having COVID-19 history (S2 Table).’ 

We added on page 7 line 6:

‘Similar results were also obtained when restricting the study period from July 15 2021, to September 30, 2021, only (Table 3) or when excluding pairs with at least one individual having COVID-19 history (S2 Table)’ 

5) The manuscript doesn’t have strengths and limitations part in the discussion. I would think that the manuscript would benefit from this. I would argue that one of the weakness is that all Covid-19 hospitalized did not have positive SARS-CoV-2 sample in the registers. One strength would be that you have quite comprehensive registers in use.

a. I’m sorry that I did not comment on this in the first revision round

We added the following paragraph in the discussion section:

‘The SNDS is a comprehensive claims database that allowed us to analyse vaccine effectiveness for individuals from the whole population of the French overseas territories benefiting from a 2-dose vaccination schedule, thus limiting selection bias. We focused on patients requiring admission specifically for SARS-CoV-2 infection, and due to the invoicing nature of the codifications recorded by public and private healthcare establishments, diagnostic errors are likely to be rare. Given the predominance of the Delta variant over the period analyzed, our estimates may not apply to periods of high circulation of other variants, such as Omicron. We may have a misclassification bias due to differential health care seeking behavior between vaccinated individuals and unvaccinated controls. This may lead to a potential underestimation of illness among unvaccinated individuals and then an underestimation of vaccine effectiveness. Health care seeking behavior may also be considered as a confounding factor: people with less use of healthcare would be less vaccinated and hospitalized for covid-19. However, this would have a limited impact due to the severity of the endpoint, and would then lead to an underestimation of vaccine effectiveness. Our study may also have been affected by other residual confounding due to differences between vaccinated individuals and unvaccinated controls after matching and adjustment for a high range of comorbidities that were shown to be associated with COVID-19-related hospitalization.’

The fact that not all Covid-19 hospitalized patients had a positive SARS-CoV-2 sample in the registers is not really a limitation. We focused on patients requiring admission specifically for SARS-CoV-2 infection, and due to the invoicing nature of the codifications recorded by public and private healthcare establishments, diagnostic errors are likely to be rare. Codification errors concerning the reason for hospitalization admission would indeed constitute a fraud from the healthcare establishment, which could then suffer heavy financial penalties. The control of these important financial flows by the paying agencies then contributes to the good quality of the coding.

---

## [Decision Letter · Decision Letter 2]

26 Aug 2022

Vaccine effectiveness against severe COVID-19 outcomes within the French overseas territories: a cohort study of 2-doses vaccinated individuals matched to unvaccinated ones followed up until September 2021 and based on the National Health Data System

PONE-D-22-10419R2

Dear Dr. Semenzato,

We’re pleased to inform you that your manuscript has been judged scientifically suitable for publication and will be formally accepted for publication once it meets all outstanding technical requirements.

Kind regards,

Joël Mossong, PhD

Academic Editor

PLOS ONE

Additional Editor Comments (optional):

Reviewers' comments:

Reviewer's Responses to Questions

**Comments to the Author**

1. If the authors have adequately addressed your comments raised in a previous round of review and you feel that this manuscript is now acceptable for publication, you may indicate that here to bypass the “Comments to the Author” section, enter your conflict of interest statement in the “Confidential to Editor” section, and submit your "Accept" recommendation.

Reviewer #1: All comments have been addressed

Reviewer #2: All comments have been addressed

2. Is the manuscript technically sound, and do the data support the conclusions?

Reviewer #1: Yes

Reviewer #2: Yes

3. Has the statistical analysis been performed appropriately and rigorously? 

Reviewer #1: Yes

Reviewer #2: Yes

4. Have the authors made all data underlying the findings in their manuscript fully available?

Reviewer #1: No

Reviewer #2: Yes

5. Is the manuscript presented in an intelligible fashion and written in standard English?

Reviewer #1: Yes

Reviewer #2: Yes

6. Review Comments to the Author

Reviewer #1: The authors have nicely revised the manuscript and I therefore recommend it for publication.

Before publication the authors may still consider the wording of the abstract's last methods sentence ("Analyses were ALSO stratified according to the number of comorbidities"). The "also" may not be needed or it would fit better if also covariate adjustment would be mentioned.

Reviewer #2: Dear authors and editors,

The authors have answered to my questions very well. I have just minor editing suggestions:

1) S2, S5 and S6 tables have "Region" column, although there aren't any stratification according to it. Therefore, it is likely unnecessary. Or have the authors wanted to point out that all overseas territories are included to the table?

2) Discussion pages 16 (in pdf file):

- "due to the invoicing nature of the codifications recorded by public and private healthcare establishments, diagnostic errors are likely to be rare." What does this mean? I don't understand what does this mean.

7. PLOS authors have the option to publish the peer review history of their article (what does this mean?). If published, this will include your full peer review and any attached files.

Reviewer #1: No

Reviewer #2: No

---

## [Editor Report · Acceptance letter]

2 Sep 2022

PONE-D-22-10419R2 

Vaccine effectiveness against severe COVID-19 outcomes within the French overseas territories: a cohort study of 2-doses... 

Dear Dr. Semenzato:

I'm pleased to inform you that your manuscript has been deemed suitable for publication in PLOS ONE. Congratulations! Your manuscript is now with our production department. 

Kind regards, 

on behalf of

Dr. Joël Mossong 

Academic Editor

PLOS ONE